# *Actinomyces* spp. Prosthetic Vascular Graft Infection (PVGI): A Multicenter Case-Series and Narrative Review of the Literature

**DOI:** 10.3390/microorganisms11122931

**Published:** 2023-12-06

**Authors:** Giovanni Del Fabro, Sara Volpi, Benedetta Fumarola, Manuela Migliorati, Davide Bertelli, Liana Signorini, Alberto Matteelli, Marianna Meschiari

**Affiliations:** 1Department of Infectious Diseases, Spedali Civili di Brescia, University of Brescia, 25123 Brescia, Italy; 2Clinic of Infectious Diseases, University Hospital of Modena, 41124 Modena, Italy

**Keywords:** actinomyces, prosthetic vascular graft infection, vascular surgery, aorto-enteric fistula

## Abstract

Background: Actinomycosis represents a challenging and under-reported complication of vascular surgery. Optimal management of *Actinomyces* spp. prosthetic vascular graft infection (PVGI) is highly uncertain because of the paucity of reports on this disease. Methods: We conducted a retrospective case-series of *Actinomyces*-PVGI that occurred in the last five years in two major university hospitals in northern Italy. We searched for previously published cases in the scientific literature. Results: We report five original cases of *Actinomyces* spp. prosthetic vascular graft infection following aortic aneurysm repair. Our literature review retrieved eight similar cases. Most patients were immunocompetent males. Most infections were polymicrobial (11/13 cases), with a prevalence of *A. odontolyticus* involvement (3/13 cases were associated with. *Salmonella* spp. infection). All cases had a late presentation (≥4 months from graft placement), with 61% associated with an aorto-enteric fistula. All patients received antibiotic therapy, but the duration was highly heterogeneous (from two weeks to life-long antibiotics). The patients without surgical revision experienced septic recurrences (2/13), permanent dysfunction (1/13), or a fatal outcome (2/13), while of the remainder who underwent vascular graft explant, six recovered completely and one developed a periprosthetic abscess. In two cases follow-up was not available. Conclusions: This case-series aims to raise the diagnostic suspicion and to describe the current management of *Actinomyces*-PVGIs. We highlight a high heterogeneity in antibiotic duration, choice of the antibiotic regimen, and surgical management. Higher reporting rate is advisable to produce better evidence and optimize management of this rare complication of vascular surgery.

## 1. Introduction

*Actinomyces* is an anaerobic Gram-positive rod residing on mucosal surfaces of the gastrointestinal, respiratory, and urogenital tracts. It can cause endogenous infections, of which the most common forms are orocervical, thoracic, abdominal, and pelvic actinomycosis [1,2]. However, actinomycosis can virtually affect any body site. Some examples of unusual presentations include gallbladder infection, pericarditis, infection of the penile shaft, infected lymphocele, and infections of prosthetic joints [3,4,5,6]. *Actinomyces* spp. bloodstream infections (BSI), eventually causing endocarditis, have been reported [7,8]. *Actinomyces* has never been well-characterized as a pathogen complicating vascular surgery procedures.

In the present study, we report five cases of prosthetic vascular graft infection (PVGI) caused by *Actinomyces* spp., and we provide a review of other cases in the literature. The aim of this narrative review is to describe the presentation, microbiological characteristics, treatment management, and outcomes of *Actinomyces*-PVGI. 

## 2. Materials and Methods

A retrospective data collection was conducted to identify all adult patients with a diagnosis of PVGI caused by *Actinomyces* spp., occurring from 1 January 2018 to 1 January 2023 in two tertiary care centers in northern Italy (University Hospitals of Brescia and Modena). Inclusion criteria included age ≥18 years, a diagnosis of PVGI based on MAGIC criteria (at least one major clinical/radiological/laboratory criterion, plus any other criterion from another category, see [9]) associated with the detection of *Actinomyces* spp. in intra-operative samples or in blood-cultures. Exclusion criteria included suspect PVGI cases that did not meet MAGIC criteria for diagnosis, lack of consent for publication, absence of microbiological diagnosis, and patients lost to follow-up at 3-months. Data relative to primary graft implant, infection signs, microorganisms involved, surgical and medical treatment, and outcomes were retrieved from electronic medical records. The study was conducted in accordance with the Declaration of Helsinki, and all subjects (or next-of-kin for dead patients) gave their informed consent for anonymous publication of data. Because of the retrospective nature of data collection, ethics approval was waived for the study. A narrative review of the English, Italian and French-language literature on *Actinomyces-PVGI* was conducted in PubMed/NCBI, Google Scholar and other similar databases. The primary aim of this review was to identify previous cases of PVGI by *Actinomyces* spp. in order to describe the number of previously published cases. Secondary objectives were to identify the patients’ characteristics and their clinical management. Key search terms were “actinomyces” OR “actinomycosis” cross-referenced with “vascular infection” OR “vascular graft” OR “vascular prosthesis” OR “vascular surgery”. Inclusion criteria included adult patients with aortic graft infection, the presence of detailed description of microbiological results in the report, and a definite microbiological diagnosis (*Actinomyces* spp. identification in the intraoperative samples or in blood cultures). Reports not providing details on patient characteristics, clinical management, and outcome were included only if they provided sufficient details on diagnostic definition of *Actinomyces*-PVGI. Studies regarding animals or experimental models, duplicate papers and cases without a definite microbiological diagnosis were excluded. The remaining papers were reviewed, and all papers that did not specifically comprise cases of PVGI (as indicated in the title or abstract) were excluded. After applying these criteria, 14 papers were included in the literature review. The results of this search are listed in the Results Section 3.2 and commented on in Section 4.

## 3. Results

### 3.1. Case Series

Case 1: A 73-year-old male patient underwent endovascular repair (EVAR) and sigmoidectomy for abdominal aortic aneurysm (AAA) rupture with intestinal ischemia in August 2021. Seven months later, he was admitted for abdominal pain and hemoptysis. A CT-scan revealed a periprosthetic fluid collection with air bubbles and an aorto-enteric fistula (AEF). Graft replacement and partial ileal resection were performed. The intraoperative samples isolated *Actinomyces odontolyticus* and *Candida albicans*. Two weeks of meropenem plus micafungin were followed by eleven months of fluconazole and long-term amoxicillin. After seven months, a CT-scan showed a new appearance of periprosthetic left psoas abscess. The patient refused any new intervention, therefore suppressive antibiotic therapy was established. At the two-year follow-up, the patient was clinically and radiologically stable, with no signs of systemic infection. 

Case 2: A 81-year-old male patient underwent an EVAR procedure for AAA by *Salmonella* spp. in January 2021. Six months later, he was readmitted for fever and lumbar pain. A PET/CT-scan revealed a periprosthetic collection with increased glucidic uptake. Debridement and graft replacement were performed. The intraoperative cultures identified *Salmonella* spp. and *A. odontolyticus*. Twelve weeks of intravenous β-lactams were followed by 12 months of oral amoxicillin. At the two-year follow-up, the patient had no relapse.

Case 3: A 77-year-old male patient underwent aorto-bisiliac prosthesis placement for AAA in 2014. Three years later, it was complicated by endoleak and AEF, requiring aneurysmatic-sac embolization and surgical repair. In May 2018, he presented with fever, lumbar pain and anemia. CT-angiography showed PVGI and a new AEF. Blood cultures isolated *A. odontolyticus*. Due to his poor health status, a conservative management with oral amoxicillin was attempted. The patient died from an aneurysmatic rupture 37 days after the onset of the infection.

Case 4: In May 2018, a 69-year-old man underwent an emergency EVAR because of a AAA rupture. The rupture was suspected to be infectious based on its uptake at CT-scan, although without any microbiological isolate. Three months later, he was readmitted for sepsis due to *A. odontolyticus* and *S. anginosus* bloodstream infection (BSI). *Salmonella* spp. grew from stool cultures. A PET/CT-scan showed increased uptake of the endovascular prosthesis. PVGI was confirmed at open surgery and prosthesis explant was performed. Given the negativity of all the intraoperative samples, antibiotic therapy was stopped two weeks after surgery. At the three-year follow-up, the patient had no relapse, with significant reduction of the uptake at the PET/CT-scan.

Case 5: A 76-year-old woman underwent an EVAR procedure for AAA by *Salmonella* spp. in December 2020. The course was complicated by abscessualization of the aneurysmatic-sac and vertebral osteomyelitis, requiring surgical drainage, vertebral stabilization, and twelve weeks of antibiotic therapy. Four months later, she was readmitted for spinal implant infection (requiring implant removal) and polymicrobial BSIs by *S. aureus*, *E. coli*, and *Actinomyces* spp. A PET/CT-scan showed an increased uptake of the vascular prosthesis, suspicious for PVGI. Due to her poor health status, explant of the vascular graft was infeasible. Long-term antibiotic treatment was started. The patient died in an end-of-life-residency seven months later. 

All cases are summarized in Table 1 and Table 2. 

### 3.2. Literature Review

We could find only six detailed reports of PVGI by *Actinomyces* spp. specifically involving aortic grafts [10,11,12,13,14,15], with five in English and one in French. Two distinct cases of PVGI caused by *Actinomyces odontolyticus* by the same author were identified in a study focusing on performance of graft culture with and without sonication [16] and in one study focusing on vascular infection by *Coxiella burnetii* [17]. Even if no complete clinical details were reported, we retained the two cases as valuable. Four additional cases were found in retrospective cohort studies of patients with vascular graft complicated by an AEF [18] or infection [19,20,21], but they were not included because no specific information was provided.

Two further cases of actinomycosis (identification of *Actinomyces oris* and *israelii*) associated with percutaneous coronary intervention (performed four months before) were retrieved from the literature search [22,23]. Because of the different anatomical setting and presentation, these two reports were not included in the list of PVGI cases caused by *Actinomyces* spp.

The significant findings from the literature search [10,11,12,13,14,15,16,17] are summed-up in Table 3 and Table 4. All eight cases occurred in male patients with an age ranging from 54 to 79 years. Seven cases (88%) were associated with AEF (in the case of Howgego et al., the AEF was primary) [10,11,12,13,14,15,16], but in the remaining case [17] an aorto-duodenal “close contact” was reported. In all cases, the presentation was delayed (>4 months after implant), with a time from graft implant to *Actinomyces* spp. infection ranging from >4 months to eight years. Polymicrobial infection was the most common (7/8 cases, 88%). Only in one case a monomicrobial infection by *Actinomyces* spp. was described [16]. However, this episode was preceded two years before by a septic episode with microbiological identification of *Streptococcus milleri*, *Eikenella corrodens*, and *Bacteroides* sp. 

The management was heterogeneous. In three cases, a conservative approach was attempted [10,13,15]. In the case presented by Delarbre, this approach was initially attempted, with no success. After failure, surgical *debridement* with complete removal of the infected graft was performed [10]. *Debridement* with complete vascular prosthesis removal and AEF repair were performed in four cases [10,11,12,14]. In two cases, the management is unknown [16,17]. Antibiotic duration after surgical procedure ranged from two weeks [10] to long-term therapy [12,13,15]. Patients treated with surgery [10,11,14] were reported to have a full recovery (in the case presented by Lane, there was no detailed follow-up after discharge [12]). Patients treated conservatively [13,15] suffered recurrent episodes of infection [13] or permanent dysfunction [15].

## 4. Discussion

The most common microorganisms involved in PVGI are *S. aureus*, coagulase-negative staphylococci, and Gram-negative bacilli (especially *E. coli* and *P. aeruginosa*) [24,25]. Only eight sufficiently detailed cases of *Actinomyces-PVGI* were previously reported in the literature [10,11,12,13,14,15,16,17]. So far, this case-series constitutes the largest case-series about *Actinomyces*-PVGI specifically involving aortic grafts.

*Actinomyces* spp. are an emerging cause of endocarditis [26], implying an endothelial adherence capacity that may also be involved in PVGI pathogenesis. The implanted prosthesis itself is a predisposing factor for actinomycosis due to its ability to form biofilm [1]. Biofilm-associated infections constitute a unique setting for infection development: even a low inoculum of a low-grade pathogen is sufficient to cause an implant-associated infection [27,28]. Our case-series shows that, together with intrauterine contraceptive devices [29], arthroplasties [6], mammary prosthesis [30], and intraocular implants [31], even vascular grafts appear to be susceptible to *Actinomyces* invasion. 

Local tissue damage, poor dental status, trauma, or surgery serves as an entry site for *Actinomyces* [2]. PVGI may arise through three pathways: (1) contamination during graft implant procedure, (2) intestinal bacterial translocation, especially in the case of pre-existing AEF, and (3) BSI. The first two are likely the most common. Overall, considering our patients and the eight previously published cases, 61% (8/13) of patients had a secondary AEF [10,11,12,13,14,16]. AEF formation can alternatively be the cause and the effect of graft infection [32]. Intriguingly, *Actinomyces* infection is a recognized cause of primary AEF, often occurring in the duodenum [33,34]. Microbiological analysis of arterial aneurysms shows that bacteria can be detected in one quarter of the aneurysm walls, and anaerobic microorganisms (including *Actinomyces* spp.) are present in 71% of cases [35]. However, when PVGI is already present, it is almost impossible to understand if AEF is primary (causing the PVGI) or secondary to graft infection. In our case-series, three of five cases underwent an EVAR procedure because of AAA related to *Salmonella* spp. infection (case 2, 4, and 5). It is difficult to conclude if PVGI occurred as a consequence of a persistent infection of the aneurysmatic wall or as a consequence of bacterial translocation occurring after the graft implantation.

*Actinomyces* is usually involved in polymicrobial infections [1], as confirmed by our case-series and the literature review (11/13 cases, 85%) [10,11,12,14,15,16,17]. The only two cases of apparently monomicrobial infection (case 3 and Hansen et al. [13]) were not surgically treated: the absence of intraoperative samples may explain the missing identification of co-pathogens. The pathogenetic hypothesis of intestinal translocation appears to be even more relevant considering that more than half of the cases were caused by the species *A. odontolyticus* (8/13 cases, 61%). *A. odontolyticus* is a resident bacterium of the oral cavity (being the main responsible for tooth biofilm formation), pharynx, distal esophagus, and distal urinary tract [1,36]. This pathogen has been reported to cause orocervicofacial [37,38,39], laryngeal [40,41], thoracic [42,43], renal [44,45], and pelvic actinomycosis [46,47], infection of the penile shaft [4], peritonitis [48], cholecystitis [49], liver abscesses [50,51], chronic conjunctivitis [52], spinal and brain abscesses (including meningitis) [53,54,55], foot, finger, arm abscesses [56,57,58], and other soft tissues involvement, such as cutaneous abscesses [59,60], osteomyelitis [61,62], purulent pericarditis [63,64], endocarditis [65], and BSI [66,67,68,69]. Looking at implant-associated infections with the contribution of *A. odontolyticus*, besides PVGI caused by *A. odontolyticus* [10,14,16,17], reports exist on endocarditis related to implantable cardioverter defibrillator [70], IUCD-associated pelvic actinomycosis [71,72], and dental implant fixtures infection [73]. Dental or oral conditions [54,63,64], intravenous drug abuse [60,67], toothpick skin puncture [56], surgery of the gastrointestinal tract [62], laryngeal injections [41], and urologic procedures [4,45,74] have been implicated in *A. odontolyticus* infection.

Considering host predisposing factors, the male gender is known to be associated with actinomycosis [1]. That was confirmed in our case-series and review, since 80% (4/5) of our patients and 100% of the previously reported cases were males [10,11,12,13,14,15,16,17]. In our case-series the mean age was 75.2 years (range: 69–81 years), quite in line with previous cases of *Actinomyces*-PVGI [10,11,12,13,14,15,17]. Dental status is associated with actinomycosis and dental procedures are known to produce transient *Actinomyces* spp. bacteriemia [1,75]. In the case of Delarbre et al., periodontal disease was documented [10]. Detailed information on dental status is lacking in our case-series, as it is not routinely reported on clinical charts. Dampened immune status has been associated with actinomycosis [2], but in our cohort it was not possible to detect significant immunosuppressive conditions. Only case 4 had a cardias adenocarcinoma treated with surgery, chemotherapy and radiotherapy, predisposing him to mucosal damage. Also chronic kidney disease (CKD), present in case 3, has been associated with increased permeability of gastrointestinal mucosa, together with immune dysfunction [76,77]. A discrete number of single-case reports of actinomycosis in patients with CKD, including end-renal disease, have been published, suggesting a possible association between the two diseases [10,78,79].

The diagnosis of actinomycosis requires a high index of suspicion. Culture identification is not granted, as no good selective media exists, and anaerobic conditions together with long incubation are required [80,81]. Case-series on actinomycosis report that more than 40% cases are culture negative [82]. No special media is required, but it has been suggested that semi-selective media may increase isolation rates of *Actinomyces* when more rapidly growing organisms are also present [83]. Still, the polymicrobial nature of infection, mistakes in sample handling and transport (anaerobic conditions not respected), or too short incubation (at least 10 days is advisable) make it difficult to identify this slow-growing germ [80,81]. For this reason, historically, most cases of actinomycosis were identified via histopathological examination [80]. A typical finding is a granulomatous infection characterized by the presence of sulfur granules with Gram-positive filamentous branching bacteria at the periphery [2]. Carrara et al., showed the added value of histopathology compared to traditional cultures in the diagnosis of pelvic actinomycosis [84]. However, histology is not routinely performed during revisions of vascular surgery, likely contributing to under-diagnose *Actinomyces*-PVGI. Polymerase chain reaction (PCR) has been shown to be more sensitive than microbiological cultures in the setting of oral and pelvic actinomycosis [85,86]. Two recent studies about the microbiological identification in PVGI of other etiologies showed promising results of PCR techniques and sonication [16,87]. Those tools may soon become a diagnostic standard for vascular graft-associated infections, leading to increased recognition of underreported species. With the spread of molecular techniques based on PCR, such as 16s rRNA identification, and mass spectrometry techniques, including MALDI-TOF, it is expected that the diagnostic sensitivity of *Actinomyces* spp. implant-associated infection will improve, and so will the reporting rate [81,85,86].

In general, PVGI of any cause requires prosthesis explant, extensive debridement, and fistula repair to achieve infection eradication [88]. Antibiotics alone are at increased risk of clinical failure [24]. Indeed, when a conservative approach was attempted in *Actinomyces-PVGI* (case 3, case 5, the cases reported by Hansen [13] and Howgego [15], and the first approach to the case reported by Delarbre [10]), an unfavorable outcome was experienced (septic recurrences or uncontrolled infection in two cases [10,13], permanent dysfunction in one case [15], and a fatal outcome in two patients). Of the seven patients treated with graft explant, six recovered completely (case 2, case 4, the cases reported by Delarbre [10], Bush [11], Lane [12] and Blank [14]) and only one lately suffered of secondary psoas abscess (case 1). Following the surgical explant of the infected graft, vascular functionality was achieved through graft replacement (four cases; see cases 1, 2, 4, and the case by Delarbre [10]) or with a bypass (three cases, see cases [11,12,14]). Due to the small number of cases and the number of factors that influence the choice, it is not possible to assess which reconstruction type is preferable. A recent meta-analysis on PVGI of any cause aimed to assess and compare the effects of surgical and medical interventions, but it was inconclusive due to the lack of good evidences on this topic [89]. 

In our case-series, the length of antimicrobial therapy ranged from five weeks (case 4) to long-life suppression (case 1). The heterogeneity in treatment duration is also evident from the literature review (ranging from two weeks to life-long suppression). This variety is explainable by the lack of consensus on optimal antimicrobial treatment duration for both PVGI and *Actinomyces* infections [1,24]. Recommended duration of antibiotics in general PVGI is based on expert opinions, and American and European guidelines suggest a minimum of 2–4 weeks of intravenous antibiotics followed by oral therapy for a variable duration (2 weeks to life-long) [24,90]. In the case of graft replacement and adequate source control, European guidelines indicate that a total duration of 4–6 weeks of antimicrobial therapy might be sufficient [24]. *Actinomyces* spp. are usually susceptible to unprotected penicillins, and traditionally, they were treated with an “intensive phase” of intravenous antibiotics (2–6 weeks) followed by oral amoxicillin for at least 6 months [1]. More recent data, however, showed excellent results with shorter courses of antibiotics when associated with surgery [91,92,93]. It has been suggested that antimicrobial duration in actinomycosis infection should be tailored to the clinical and radiological response and the initial burden of disease [93], but the absence of clear guidelines makes it difficult to apply shorter antimicrobial courses in clinical routine. Looking at previously reported cases of *Actinomyces*-PVGI, in three cases (all associated with the surgical removal of the infected graft) postoperative antibiotics were given for less than two months, with a good final outcome [10,11,14]. Sound evidence addressing *Actinomyces*-PVGI is lacking, but it seems reasonable to conclude that antibiotic therapy might be safely shortened when adequate source control is achieved, in line with European guidelines for graft infections [24]. 

Besides duration, the choice of the antibiotic molecule deserves some consideration. Penicillins are considered the drug of choice for historical reasons, but there are no randomized control studies evaluating the efficacy of alternative regimens effective in vitro [94,95,96,97,98]. Several reports showed effective cure of actinomycosis with ceftriaxone, carbapenems, macrolides, and doxycycline [14,91,99,100,101]. Tetracyclines represent an attractive oral option in actinomycosis because of their good bioavailability and tissue penetration. They have been used with clinical success, and due to their favorable pharmacokinetic profile, they also appear to be a valuable option in *Actinomyces*-PVGI [14]. However, tetracyclines resistance (up to 30% of isolates) has been reported, so they should be used only with documented susceptibility [94,97,102]. A major limitation affecting the choice of therapy is that identification by culture is not easy and, even if available, susceptibility tests for anaerobes are not routinely performed in many laboratories, with interpretation being limited to CLSI breakpoints. The E-test method is not the gold standard for anaerobic bacteria, but it is more practical and cost-effective then the agar dilution method. In a recent work on oral isolates, a strain of *A. odontolyticus* resistant to benzylpenicillin, meropenem, moxifloxacin, and daptomycin was detected [103]. Surveillance data and clinical studies raise concerns about antimicrobial resistance trends in anaerobic infections [104]. Thus, routine susceptibility testing for anaerobes should be strongly encouraged to guide antimicrobial therapy.

Given the many points of uncertainty regarding *Actinomyces*-PVGI, a long-term follow-up with close surveillance is recommended, including laboratory tests and imaging. In particular, CT and PET/CT scans can support clinicians in managing these cases [24]. In case 1, the CT-scan identified an otherwise asymptomatic periprosthetic abscess of the psoas muscle, confirming the importance of appropriate radiological follow-up.

Our results may not be generalizable due to the small number of cases, the heterogeneity of management, and the retrospective nature of data collection (with possible selection bias). Nevertheless, this descriptive study represents the largest case-series published so far, and it may increase awareness of an under-reported complication of vascular surgery. *Actinomyces*-PVGI following vascular graft replacement must be suspected, even in immunocompetent patients. Appropriate identification methods should be implemented because a combined approach of antibiotic therapy and surgery improves the final outcome. The optimal antimicrobial duration is currently unknown. We conclude with a call for increasing *Actinomyces*-PVGI reports in order to produce better evidence for the appropriate management of this disease.

## Figures and Tables

**Table 1 microorganisms-11-02931-t001:** Demographic and clinical characteristics of *Actinomyces*-PVGI cases occurred in the last five years in the University Hospitals of Brescia and Modena.

N	Age (y), Sex (M/F)	Comorbidities	Aortic Prosthesis Implant		Clinical Picture	
			Reason for Primary Vascular Implant (Type of Procedure)	Time from Implant to Presentation	Presentation	Aorto-Enteric Fistula
1	73, M	Appendectomy, previous AKI on CVVH	AAA rupture with intestinal ischemia (EVAR and sigmoidectomy)	7 m	Abdominal pain and hemoptysis	Yes (ileum)
2	81, M	Hypertension, atrial fibrillation, previous SARS-CoV-2 pneumonia	infectious AAA due to *Salmonella* spp. (EVAR)	6 m	Fever and lumbar pain	Not identified
3	77, M	Dialysis	AAA (aorto-biiliac prosthesis, procedure unknown)	4 y	Fever, lumbar pain and intestinal bleeding	Yes (duodenum)
4	69, M	Cardias adenocarcinoma treated surgically + chemotherapy + radiotherapy, complicated with esophageal fistula surgically repaired (10 y before), previous lymphoma	infectious AAA rupture secondary to gastrointestinal infection (EVAR and PTA of left common iliac artery)	4 m	Low-grade fever and septic shock	Not identified
5	76, F	Mammary carcinoma treated surgically (4 y), left TKA, intestinal subocclusion (2 y), COPD, hypertensive cardiomyopathy, amoxicillin allergy (rash)	infectious AAA rupture due to *Salmonella* spp. (EVAR) with subsequent abscessualization of aneurysmatic sac and vertebral osteomyelitis requiring surgical drainage and spine stabilization	10 m	*S. aureus* vertebral implant infection with wound dehiscence later complicated with polymicrobial BSIs	Not identified

Abbreviations: N: number; M: male; F: female; AKI: acute kidney injury; CVVH: continuous veno–venous hemofiltration; COPD: chronic obstructive pulmonary disease; PTA: percutaneous transluminal angioplasty; TKA: total knee arthroplasty; AAA: abdominal aortic aneurysm; EVAR: endovascular repair; d: days; w: weeks; m: months; y: years.

**Table 2 microorganisms-11-02931-t002:** Microbiological characteristics, treatment, and outcomes of *Actinomyces*-PVGI cases that occurred in the last five years at the University Hospitals of Brescia and Modena.

N	Microbiological Results	Treatment		Outcome
		Antibiotic Therapy	Surgery	
1	*A. odontolyticus* (I) *C. albicans* (I)	iv: meropenem + micafungin (2 w) then os: amoxicillin (long-term) + fluconazole (11 m)	Explant and substitution	Periprosthetic abscess of the psoas muscle (2 y f-up)
2	*A. odontolyticus* (I) *Salmonella* spp. (I)	iv: ceftriaxone + ampicillin (12 w) then os: amoxicillin (10 m)	Explant and substitution	Recovered (2 y f-up)
3	*A. odontolyticus* (B)	os: amoxicillin (long-term)	Not performed	Death (37 d)
4	*A. odontolyticus* (B) *S. anginosus* (B) *Salmonella* spp. (S)	iv: daptomycin + amoxi/clav. (3 w) then os: amoxiclav. (15 d)	Explant and bypass	Recovered (3 y f-up)
5	*Actinomyces* spp. (B) *S. aureus* (B) *E. coli* (B)	iv: meropenem (2 w) then os: cotrimoxazole + rifampin (long-term)	Not performed	Death (7 m)

Abbreviations: d: days; m: months; y: years; B: blood cultures; I: intraoperative; S: stool cultures; iv: intravenous therapy; os: oral therapy.

**Table 3 microorganisms-11-02931-t003:** Demographic and clinical characteristics of *Actinomyces*-PVGI cases retrieved from the literature review.

Case Report	Age (y), Sex (M/F)	Comorbidities	Aortic Prosthesis Implant		Clinical Picture	
			Reason for Primary Vascular Implant (Type of Procedure)	Time from Implant to Presentation	Presentation	Aorto-Enteric Fistula (AEF)
Delarbre (2007) [10]	73, M	Peripheral obliterant arteriopathy, arterial hypertension, dyslipidemia, COPD, CKD, periodontal disease	Peripheral obliterant arteriopathy (aortobiiliac prosthesis, OSR)	7 y	Fever and lumbar pain	Yes (duodenum)
Bush (2009) [11]	79, M	-	- (EVAR)	8 y	Fever and abdominal pain irradiating to the back	Yes (duodenum)
Lane (2009) [12]	69, M	-	Infrarenal AAA (EVAR)	6 m	Fever, lethargy, diarrhea	Yes (duodenum)
Hansen (2017) [13]	75, M	-	Ruptured infrarenal AAA (EVAR) complicated with graft infection (isolation of *S. milleri*, *E. corrodens*, *Bacteroides* sp.) at 2 y, treated with graft revision (without explant), AEF closure, antibiotic therapy (meropenem iv followed by ciprofloxacin and clindamycin for 3 m)	4 y	Fever	Yes (duodenum)
Blank (2017) [14]	54, M	Hypertension, hyperlipidemia, asthma, diverticulitis	Acute limb ischemia (open aortobifemoral bypass)	6 m	Fever, left leg pain	Yes * (sigmoid colon)
Puges (2018) [16]	-	-	-	>4 m	-	Yes (-)
Howgego (2021) [15]	68, M	-	Ruptured AAA with primary AEF (EVAR) complicated with graft infection at 3 m, treated with graft revision (without explant) and long-term amoxiclavulanate	1 y	Sepsis	No
Puges (2021) [17]	78, M	-	-	6 y	Fever, back pain and acute respiratory failure	No ^

Abbreviations: CKD: chronic kidney disease; COPD: chronic obstructive pulmonary disease; EVAR: endovascular repair; d: days; m: months; y: years; OSR: open surgical repair. * Unclear if the AEF was secondary to a technical error during graft placement. ^ Close aortoduodenal contact.

**Table 4 microorganisms-11-02931-t004:** Microbiological characteristics, treatment, and outcomes of *Actinomyces*-PVGI cases retrieved from the literature review.

Case Report	Microbiological Results	Treatment		Outcome
		Antibiotic Therapy	Surgery	
Delarbre (2007) [10]	*A. odontolyticus* (B) *E. coli* (B, I) *E. faecium* (I) *Candida albicans* (I)	First (conservative) iv: amoxicillin iv (3 w) + gentamicin (10 d); then os: amoxicillin (8 m)	First: Not performed	First: failure;
		Second: iv: imipenem + amikacin + fluconazole (2 w)	Second: Explant and graft replacement	Second: Recovered (6 y f-up)
Bush (2009) [11]	*Actinomyces* spp. (B) *S. constellatus* (B, I) *P. melaninogenica* (B) *S. lugdunensis* (I)	iv: ampic./sulbactam (8 w)	Explant and axillo-bifemoral graft bypass	Recovered (2 m f-up)
Lane (2009) [12]	*A. israelii* (I) *B. fragilis* (I)	Unknown (“suppressive antibiotics”)	Explant and axillo-bifemoral graft bypass	Discharged (-)
Hansen (2017) [13]	*Actinomyces* spp. (B)	Unknown (“suppressive antibiotics”)	Not performed	Recurrent sepsis and digestive bleeding (alive at 12 y f.up)
Blank (2017) [14]	*A. odontolyticus* (I) *S. epidermidis* (I) *S. anginosus* (I)	iv: tigecycline (6 w) then os: doxycicline (6 w)	Explant and axillary-femoral bypass, colectomy	Recovered (1 y f-up)
Puges (2018) [16]	*A. odontolyticus* (I) *K. pneumoniae* (I) *V. parvula* (I) *C. albicans* (I) *C. tropicalis* (I)	-	-	-
Howgego (2021) [15]	*Actinomyces* spp. (B) *E. faecium* (B)	iv: meropenem (later ertapenem) + vancomycin (later teicoplanin) then: long-term amoxiclavulanate	Not performed	Discharged, permanent kidney dysfunction (-)
Puges (2021) [17]	*A. odontolyticus* (I) *S. anginosus* (I) *S. oralis* (I, B) *Coxiella burnetii* (serology and PCR)	Unknown (no *Coxiella* treatment)	-	Died at 37 d

Abbreviations: d: days; m: months; y: years; B: blood cultures; I: intraoperative; iv: intravenous therapy; os: oral therapy.

## Data Availability

Data available on request from corresponding author due to privacy restrictions.

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
