# Peer review of "Actinomyces spp. Prosthetic Vascular Graft Infection (PVGI): A Multicenter Case-Series and Narrative Review of the Literature"

_microorganisms, 2023, doi:10.3390/microorganisms11122931_

Round 1

Reviewer 1 Report

Comments and Suggestions for Authors

Prosthetic vascular graft infection is rare but fatal and complicated to treat. In this report, the authors conducted a retrospective case-series of Actinomyces-PVGI that occurred in two major university hospitals and analyzed eight cases previously published in scientific literature. It is a specific report with regarding to infetion of actinomyces. It is interesting but I have several comments for the authors.

1. I would suggest to define the Actinomyces spp.

2. As far as I know, cases 2 and 5, may including case 4 can be diagnosed with mycotic AAAs (or infectious AAA). However, case 1 and 3 were diagnosed with vascular graft infection. Although both may have similar treatment, they are different entities associated with different etiologies and prognosis. They are usually investigated separately. Please clarify.

3. For surgical treatment, which reconstruction type, in-situ or bypass, would be preferred?

Comments on the Quality of English Language

Good

Author Response

COVER LETTER

Hereby we provide the answers to the reviewers. We are thankful to the reviewers for the time dedicated and the useful comments. We revised the manuscript with some major changes. We added details on ethical approval (METHODS) that was missed in the draft and we changed the classification from “systematic” to “narrative review”, because our search did not satisfy all the criteria for a systematic review. We will answer point by point to the interesting comments listed.

REVIEWER 1

Answer:
Thank you very much for the time dedicated for your review. I will answer point by point to the interesting comments listed.
1. I would suggest to define the Actinomyces spp.

ANSWER: Do you mean to define the Actinomyces sp. by a microbiological point of view? In the introduction, lines 31-32 is reported “Actinomyces is an anaerobic Gram-positive rod residing on mucosal surfaces of the gastrointestinal, respiratory, and urogenital tract”. The microbiological characteristics of Actinomyces are defined again in the manuscript in the Discussion. In particular a focus on anaerobic characteristics, propensity to cause polymicrobial infections, slow growth, microscopy aspects and other particularities are defined in the microbiological diagnosis section (lines 255-279). We did not write a specific “microbiological” section because we liked to put this information in a narrative way, to be more fluent and because we offer a clinical point of view rather than a laboratory point of view. However, we accept any further observation that could help to ameliorate the manuscript.

  1. As far as I know, cases 2 and 5, may including case 4 can be diagnosed with mycotic AAAs (or infectious AAA). However, case 1 and 3 were diagnosed with vascular graft infection. Although both may have similar treatment, they are different entities associated with different etiologies and prognosis. They are usually investigated separately. Please clarify.

ANSWER: Right. Case 1,3 and probably case 4 start as mycotic AAA. Those patients were treated with antibiotics for this reason and received an aortic graft implant because of aorta rupture. However their course got lately complicated with vascular graft infection (VGI), associated with Actinomyces. The temporal sequence was then: mycotic AAA-> graft placement & antibiotic therapy -> VGI. As discussed in Discussion, it is not possible to exclude that infection persisted and lately involved the graft, However none of the cases had an isolation of Actinomyces spp previous of graft placement, so this is just an hypothesis that can not be verified.

  1. For surgical treatment, which reconstruction type, in-situ or bypass, would be preferred?

ANSWER: As we reported in Discussion, guidelines recommend  “prosthesis-explant, extensive debridement and fistula repair to achieve infection eradication” (lines 258-259). This is possible to be achieved both with in situ reconstruction and bypass. The number of cases is too small to determine which reconstruction type is preferable. Also, strong biases in patient selection (fitness of the patient) and different surgical expertise make it fairly impossible to answer. We had this sentence in Discussion, lines 288-294: “Following surgical explant of the infected graft, vascular functionality can be achieved through graft replacement (four cases; see cases 1, 2, 4 and [9]) or with a bypass (three cases, see cases [10,11,13]). Due to the small number of cases and the number of factors that influence the choice, it is not possible to assess which reconstruction type is preferable.” And lthe following sentence “A recent meta-analysis on PVGI of any cause aimed to assess and compare the effects of surgical and medical interventions but it was inconclusive due to the lack of good evidences on the topic”.

Reviewer 2 Report

Comments and Suggestions for Authors

Thank you for providing me the opportunity to review this manuscript. It is interesting and provides essential information. I have some comments that could be of use:

1.     Since this work is also a systematic literature review, a small sentence mentioning some basic information about the methodology could be added in the abstract section. For example, a systematic review of studies published in Pubmed, Scopus (or whatever database was included) until date X was performed for relevant studies

2.     The methods section needs more description. What were the inclusion and exclusion criteria? Please define the exact date the study was conducted. What were the parameters that were extracted and evaluated?

3.     Ethics approval for the collection of the retrospective information is needed. This cannot be seen either in the methods section or at the end. Was there patient consent needed?

4.     Line 48: Please italicize all gender and species names throughout the manuscript

5.     Line 49: and other similar databases? Since this review is mentioned as systematic, specific details about the methodology are needed. What exactly were the databases that were searched? What is the date of the last search? Who did the literature search? What database did you use? Was it abstrackr? Rayyan? Who did the extraction of the data? Was there a consensus for any differences among different researchers? Did you search the references of the resulting references (snowball procedure)? Did you use GRADE or another tool for grading the evidence? Did you follow the PRISMA or the MOOSE guidelines? How did you manage the unavailability of any published studies? If these questions are not addressed, then this review should probably be named a narrative rather than a systematic

6.     Line 53: What do you mean by ‘definite microbiological diagnosis’? You must introduce a definition in that case

7.     Table 1: Please define sdr in the footnote

8.     Table 2: copro-cultures? Maybe you mean stool cultures

9.     While reading the results section, I cannot understand what are the primary objectives of the systematic review. I mean, there must also be inclusion and exclusion criteria for the systematic search. For example, did you include all cases regardless of the information provided, or did you have a minimum of information (like mortality) needed for a study to be included?

10.  Line 134: significative?

11.  A diagram of study inclusion (for example, the PRISMA flow diagram, if the PRISMA guidelines for systematic reviews are followed) is absolutely necessary. The reader must understand how many studies were initially screened, how many were rejected at screening, how many proceeded to the full-text analysis, how many were rejected at that stage and for what reason, how many were manually retrieved, and how many studies were finally included. These must be definitely graphically depicted

12.  Line 305: I would remove the word ‘neglected’

13.  The discussion section is interesting. However, comparing Actinomyces PVGI with PVGI by other bacteria could be helpful. This would allow the reader to understand any important differences in terms of epidemiology, clinical presentation, or complications

Comments on the Quality of English Language

Moderate changes needed

Author Response

COVER LETTER

Hereby we provide the answers to the reviewers. We are thankful to the reviewers for the time dedicated and the useful comments. We revised the manuscript with some major changes. We added details on ethical approval (METHODS) that was missed in the draft and we changed the classification from “systematic” to “narrative review”, because our search did not satisfy all the criteria for a systematic review. We will answer point by point to the interesting comments listed.

REVIEWER 2

Answer:
Thank you very much for the time dedicated for your review and the very interesting and detailed analysis. Summary of Major changes: we added a sentence that was forgotten about ethics approval in METHODS and we changed the classification from “systematic” to “narrative review”. We will answer point by point to the interesting comments listed.

  1. Since this work is also a systematic literature review, a small sentence mentioning some basic information about the methodology could be added in the abstract section. For example, a systematic review of studies published in Pubmed, Scopus (or whatever database was included) until date X was performed for relevant studies.

ANSWER: Changed to “narrative review” see point 5

  1. The methods section needs more description. What were the inclusion and exclusion criteria? Please define the exact date the study was conducted. What were the parameters that were extracted and evaluated?

 ANSWER: Changed to “narrative review” see point 5 It is not clear for us f in this point 2 are you addressing the literature review part or the retrospective data collection. Please clarify if we misunderstood.

  1. Ethics approval for the collection of the retrospective information is needed. This cannot be seen either in the methods section or at the end. Was there patient consent needed?

 ANSWER: Right, that is very important, we are sorry but it was missed in the draft revision. In our country (Italy) and in our specific institutions ethics approval is not needed for retrospective data coming from clinical practice. However, informed consent from subjects or parents must be taken. We added the sentence in METHODS (Lines 48-52): “The study was conducted in accordance with the Declaration of Helsinki, and all subjects (or next of kin for dead patients) gave their informed consent for anonymous publication of data. Because of the retrospective nature of data collection, ethics approval was waived for the study.

  1. Line 48: Please italicize all gender and species names throughout the manuscript ANSWER: Done. Thank you for the observation.
  2. Line 49: and other similar databases? Since this review is mentioned as systematic, specific details about the methodology are needed. What exactly were the databases that were searched? What is the date of the last search? Who did the literature search? What database did you use? Was it abstrackr? Rayyan? Who did the extraction of the data? Was there a consensus for any differences among different researchers? Did you search the references of the resulting references (snowball procedure)? Did you use GRADE or another tool for grading the evidence? Did you follow the PRISMA or the MOOSE guidelines? How did you manage the unavailability of any published studies? If these questions are not addressed, then this review should probably be named a narrative rather than a systematic

ANSWER: Your comments were very useful and detailed, as point 1, 2, 6 and 11. We realized that our search does not satisfy the criteria of a sysematic review. We think it fits greatly with a narrative review. Then, we changed the title to “narrative” review and so on we updated the manuscript. See line 52. See also point 6.

  1. Line 53: What do you mean by ‘definite microbiological diagnosis’? You must introduce a definition in that case.

 ANSWER: We added the definition (see point 9) and the inclusion and exclusion criteria in a more detailed way.  Added: “Inclusion criteria were: adult patients with aortic graft infection, the presence of detailed description of microbiological results in the report and a definite microbiological diagnosis (Actinomyces spp. identification in the intraoperative samples or in blood-cultures). Reports not providing details on patient characteristics, clinical management and outcome were included only when they provided sufficient details on diagnostic definition of Actinomyces PVGI. Studies regarding animals or experimental models, duplicate papers and cases with not definite microbiological diagnosis were excluded. he remaining papers were reviewed, and all papers that did not specifically (in the title or abstract) comprise cases of PVGI were excluded

  1. Table 1: Please define sdr in the footnote.

ANSWER:Done.

  1. Table 2: copro-cultures? Maybe you mean stool cultures.

ANSWER: Right. Changed accordingly.

  1. While reading the results section, I cannot understand what are the primary objectives of the systematic review. I mean, there must also be inclusion and exclusion criteria for the systematic search. For example, did you include all cases regardless of the information provided, or did you have a minimum of information (like mortality) needed for a study to be included?

ANSWER: We added the aim of the literature search in the METHODS section (The primary aim of this review was to identify previous cases of PVGI by Actinomyces spp, in order to describe the number of previously published cases. Secondary objectives were to identify the patients characteristics and their clinical management.”).

  1. Line 134: significative?

ANSWER: As we think that Actinomyces is an underrecognized cause of PVGI, we thought it may be useful to cite other articles reporting Actinomyces involvement in graft infections. However we eliminated the description of the case because it was redundant.

  1. A diagram of study inclusion (for example, the PRISMA flow diagram, if the PRISMA guidelines for systematic reviews are followed) is absolutely necessary. The reader must understand how many studies were initially screened, how many were rejected at screening, how many proceeded to the full-text analysis, how many were rejected at that stage and for what reason, how many were manually retrieved, and how many studies were finally included. These must be definitely graphically depicted.

ANSWER: As we changed to “narrative” review the diagram is not strictly needed. We performed a search using Pubmed and Google scholar wih the key search terms listed in the Manuscript. Titles and abstract were screened to understand if they could be relevant . We did not had problems to access full-copy of any manuscript classified as potentially relevant. All this work was manual and no automatic systems were used.

  1. Line 305: I would remove the word ‘neglected’

ANSWER: Changed accordingly

  1. The discussion section is interesting. However, comparing Actinomyces PVGI with PVGI by other bacteria could be helpful. This would allow the reader to understand any important differences in terms of epidemiology, clinical presentation, or complications. ANSWER: Thank you for the observation. Some comparisons between Actinomyces and other bacteria are implicit in Discussion, as we compare Actinomyces treatment with European guidelines for bacterial PVGI (not specifically addressing Actinomyces). However it seems useful to put it in a more explicit way the comparison between different etiologies. We added the sentence Common microorganisms involved in PVGI are  Gram-negative bacilli, S. aureus and coagulase-negative staphylococci [REF]. Only eight sufficiently detailed cases of Actinomyces PVGI  were previously reported in literature [9–16]. So far, this case-series constitutes the largest case-series about Actinomyces PVGI specifically involving aortic grafts.” (lines 183-87 ) and we modified lines 281-83: “In general, PVGI of any cause requires prosthesis-explant, extensive debridement and fistula repair to achieve infection eradicationachieved, in line with European guidelines for graft infections [86]; lines 302-309 “In our case-series, the length of antimicrobial therapy ranged from five weeks (case 4) to long-life suppression (case 1). The heterogeneity in treatment duration is also evident from the literature review (ranging from two weeks to life-long suppression). This variety is explainable by the lack of consensus on optimal antimicrobial treatment duration for both PVGI and Actinomyces infections [1, 23]. Recommended duration of antibiotics in general PVGI is based on expert opinions and American and European guidelines suggest a minimum of 2-4 weeks of intravenous antibiotics followed by oral therapy for a variable duration (2 weeks to life-long) [REF]. In the case of graft replacement and adequate source-control, European guidelines indicate that a total duration of 4-6 weeks of antimicrobial therapy might be sufficient [23]

Round 2

Reviewer 2 Report

Comments and Suggestions for Authors

The authors have addressed almost all comments. In the second comment at the first round of revisions, I meant to provide the inclusion and exclusion criteria/definition for PVGI infection in the original/retrospective part of the study.

Comments on the Quality of English Language

The manuscript has been improved. There is only one minor comment, as stated above.

Author Response

Clear. We changed accordingly as follows:

METHODS

A retrospective data collection was conducted to identify all adults-patients with a diagnosis of PVGI caused by Actinomyces spp., occurred from 1st January 2019 to 1st January 2023 in two tertiary care centers of northern Italy (University Hospitals of Brescia and Modena). Inclusion criteria were: age ≥ 18 years, a diagnosis of PVGI based on MAGIC criteria (at least one clinical/radiological/laboratoristic major criterion plus any other criterion from another category, see [9]) and detection of Actinomyces spp. in intra-operative samples or in blood-coltures. Exclusion criteria were: suspect PVGI cases that did not meet MAGIC criteria for diagnosis, no consent for publication, no microbiological diagnosis, and patients lost at 3-months follow-up. Inclusion criteria were: age ≥ 18 years, a diagnosis of PVGI based on MAGIC criteria (at least one clinical/radiological/laboratoristic major criterion plus any other criterion from another category, see [9]) associated with the detection of Actinomyces spp. in intra-operative samples or in blood-coltures. Exclusion criteria were: suspect PVGI cases that did not meet MAGIC criteria for diagnosis, no consent for publication, no microbiological diagnosis, and patients lost at 3-months follow-up. Data relative to primary graft implant, infection signs, microorganisms involved, surgical and medical treatment, and outcome were retrieved from electronic medical records.